# Characterization of Durum Wheat Resistance against Septoria Tritici Blotch under Climate Change Conditions of Increasing Temperature and CO$_2$ Concentration

Rafael Porras [1], Cristina Miguel-Rojas [1,*], Ignacio J. Lorite [2], Alejandro Pérez-de-Luque [1] and Josefina C. Sillero [1]

1    Area of Plant Breeding and Biotechnology, IFAPA Alameda del Obispo, Avda. Menéndez Pidal s/n, 14004 Córdoba, Spain; jrafael.porras@juntadeandalucia.es (R.P.); alejandro.perez.luque@juntadeandalucia.es (A.P.-d.-L.); josefinac.sillero@juntadeandalucia.es (J.C.S.)
2    Area of Natural and Forest Resources, IFAPA Alameda del Obispo, Avda. Menéndez Pidal s/n, 14004 Córdoba, Spain; ignacioj.lorite@juntadeandalucia.es
*    Correspondence: cristina.miguel@juntadeandalucia.es

**Abstract:** Wheat interactions against fungal pathogens, such as *Zymoseptoria tritici*, are affected by changes in abiotic factors resulting from global climate change. This situation demands in-depth knowledge of how predicted increases in temperature and CO$_2$ concentration ([CO$_2$]) will affect wheat—*Z. tritici* interactions, especially in durum wheat, which is mainly grown in areas considered to be hotspots of climate change. Therefore, we characterized the response of one susceptible and two resistant durum wheat accessions against *Z. tritici* under different environments in greenhouse assays, simulating the predicted conditions of elevated temperature and [CO$_2$] in the far future period of 2070–2099 for the wheat-growing region of Córdoba, Spain. The exposure of the wheat—*Z. tritici* pathosystem to elevated temperature reduced disease incidence compared with the baseline weather conditions, mainly affecting pathogen virulence, especially at the stages of host penetration and pycnidia formation and maturation. Interestingly, simultaneous exposure to elevated temperature and [CO$_2$] slightly increased *Z. tritici* leaf tissue colonization compared with elevated temperature weather conditions, although this fungal growth did not occur in comparison with baseline conditions, suggesting that temperature was the main abiotic factor modulating the response of this pathosystem, in which elevated [CO$_2$] slightly favored fungal development.

**Keywords:** *Zymoseptoria tritici*; climate change; components of resistance; fungal development; wheat resistance; plant–pathogen interactions

## 1. Introduction

Wheat is considered to be one of the most important food crops for human populations as it is consumed worldwide and provides substantial amounts of components that are essential or beneficial for health [1]. As a result of its key position in global cereal production and its elevated range of cultivation and diversity, this essential crop is constantly threatened by diverse biotic stresses during the growing season, including attacks by pest and pathogens, such as fungi, bacteria, oomycetes, viruses, nematodes, and herbivores, leading to a great constraint in wheat production worldwide [2–4]. Among these biotic stresses, plant diseases cause more than 21% of wheat losses on average [3], with fungal pathogens such as wheat rusts, blotch diseases, wheat scab, wheat blast, or powdery mildew, among others, being considered the most detrimental [5,6].

One of the most devastating wheat fungal diseases is Septoria tritici blotch (STB) disease, caused by the fungus *Zymoseptoria tritici* (Desm.) [7]. This major disease is present in all wheat-growing areas of the world and causes significant yield losses of up to 50% under conducive weather conditions in both common and durum wheat [8]. *Z. tritici*

normally develops well in temperate climates with cool, wet weather where bread wheat cultivation is significant, such as in North America [9], northern France, Germany, and the United Kingdom [10]. However, this pathogen also extends to hot dry climates such as wheat-growing regions of the Mediterranean Basin or North Africa, where durum wheat importance exceeds that of bread wheat due to its common use in the traditional Mediterranean diet [11,12], where lately the reintegration of ancient durum varieties of grain more resistant to pathogens is being tested [13]. This adaptation and speciation of *Z. tritici* to various agro-ecosystems is thanks to its high genome plasticity and diversity, and its active sexual reproduction, which accelerates its evolution [8], maintaining pathogenic fitness even when it loses accessory chromosomes [14,15]. This situation has hampered the implementation of an efficient strategy to control STB disease, limiting the efficacy of chemical control [8,16]. In fact, STB is probably the most economically important wheat disease in Europe, with an estimated ~EUR 1 billion per year in fungicide expenditure directed toward its control [16].

In addition to fungicide application, protection against STB disease has been traditionally achieved through the use of resistant wheat cultivars [17]. Thus, breeding programs have emerged as an effective, environmentally sustainable and cost-reducing measure to control this disease in comparison to fungicide control [18]. In STB, two types of resistance have been described: qualitative (race-specific) resistance is controlled by major genes with a large effect according to a gene-for-gene interaction [19,20], whereas quantitative (non-race-specific) resistance, which develops a partially resistant phenotype, is conferred by a large number of quantitative trait loci (QTL) with moderate to small effects [17]. Both types of resistance have been thoroughly studied in bread wheat, for which 22 major genes (*Stb* genes) conferring qualitative resistance, together with 167 QTLs, have been identified and mapped to date [21]. In fact, quantitative resistance plays an important role in wheat breeding against *Z. tritici*, which is durable and effective against several pathotypes of this pathogen [17]. This type of resistance has been usually evaluated through visual (subjective) quantitative scoring of *Z. tritici* lesions bearing fungal reproductive structures [22]. However, due to fungal lesions being absent from reproductive structures in some cases, quantitative resistance has also been evaluated by measuring lesions and reproductive structures separately [23–25]. Fortunately, recent automated image analysis methods have been postulated as an essential tool to precisely evaluate not only the amount of damage of a *Z. tritici* isolate to the host, but also its epidemic potential, obtaining an overall measurement of pathogen virulence [26,27]. Additionally, microscopic studies that determine fungal infection process and subsequent plant defense patterns, together with biochemical and molecular analyses, examining fungal enzymes or host defense-related genes, remarkably increase knowledge about this plant–pathogen interaction [28,29].

However, wheat—*Z. tritici* interaction would be affected by alterations in environmental conditions derived from global climate change, which is mainly characterized by increasing temperature, [$CO_2$], and drought [30]. In fact, alterations in temperature, [$CO_2$], and water regimes are thought to modify plant development and resistance pathways, on the one hand, and pathogen virulence mechanisms and life cycle, on the other hand [31], lastly influencing entire wheat—pathogen interactions [18]. In order to assess this influence, disease risk simulation studies (where crop disease models have been linked to climate projections) have been commonly developed for diverse global locations, showing diverse outputs for STB incidence across European wheat-growing areas [32,33]. Moreover, although researchers recognize that extreme weather events, which are characteristic of climate change, will have large impacts on disease severity and yield loss, its projections are still at the early stages [34].

Disease models are based on the results of experimental investigations that, thanks to the use of diverse facilities (free-air $CO_2$ enrichment systems, phytotrons, or greenhouses), assess the effect of one or several simultaneous abiotic factors, such as high temperature and elevated [$CO_2$], on plant–pathogen interactions [34]. However, there are few realistic field studies investigating the effects resulting from combining simultaneous increasing

temperature and [$CO_2$] with biotic stresses in plants [35], and, in the case of wheat, they often obtained varied results regarding different wheat—pathogen interactions [36–38]. These variations are a consequence of plants expressing a tailored physiological and molecular response when facing multiple stresses, triggering antagonistic signaling pathways of abiotic and biotic factors [35,39], such as the role of some phytohormones [40,41]. However, although some studies have evaluated the effect of increasing temperature [42–45] and [$CO_2$] [46] in wheat—*Z. tritici* interactions, none of them have assessed the effect of both abiotic factors acting simultaneously. Therefore, considering this knowledge gap, the variability in experimental assessments, and the uncertainty of disease risk simulation studies, it is likely that predicting wheat—*Z. tritici* interactions under climate change conditions would be a complex task.

Besides this situation, it is necessary, considering that the interaction of durum wheat against *Z. tritici* has been poorly investigated [28], leading to the absence of any *Stb* genes identified in durum wheat species. This lack of resistant resources against STB disease in durum wheat species implies a great risk for general wheat production, especially in wheat-growing regions in the Mediterranean Basin, such as Spain [12] or Tunisia [11], which altogether stand out for being the largest durum-wheat-producing areas in the world, with about 60% and 75% of the global durum wheat cropping areas and production, respectively [47]. Additionally, these regions are considered hotspots of climate change, where temperature warming, extreme events, and changes in precipitation regimes are likely to occur [48,49]. In this context, durum wheat cultivation would be severely affected by climate change and the *Z. tritici* pathogen, which is characterized by evolving and rapidly adapting to changing environments [8]. Therefore, there is an urgent necessity to gain knowledge about how future conditions of climate change would affect durum wheat—*Z. tritici* interactions.

The aim of this study was to characterize the response of durum wheat accessions with diverse resistance traits against STB disease under increasing temperature and [$CO_2$] weather conditions, according to climate projections for the far future period of 2070–2099 for the wheat growing region of Córdoba, Spain.

## 2. Materials and Methods

### 2.1. Plant Material

In this study, 3 durum wheat accessions (*T. turgidum* ssp. *durum*) selected from the germplasm collection used in Porras et al. [24], 'Sy Leonardo', 'LG Origen', and 'RGT Rumbadur', were evaluated against a local isolate of *Zymoseptoria tritici* under baseline (control) and climate change conditions. These accessions were commercial Spanish cultivars registered in the Spanish MAPA (Ministerio de Agricultura, Pesca y Alimentación) catalog. Additionally, they were classified according to their reactions against *Z. tritici* infection using the DS (disease severity) rating scale from 0 to 5 [20], with 'RGT Rumbadur', classified with a DS value of 2, and 'LG Origen', with a DS value of 3, considered as resistant accessions, and 'Sy Leonardo', with a DS value of 5, considered as susceptible [24].

### 2.2. Pathogen Isolation

The fungus Zymoseptoria tritici was isolated from naturally infected wheat leaves collected in Santaella, Córdoba (Spain), according to procedures of Stewart and McDonald [50], and the isolate was molecularly identified with the GeneBank accession number MZ026796 at the NCBI database [51]. The fungus isolation, purification, multiplication, and conservation procedures were previously described by Porras et al. [24]. Fungal stocks were stored as microconidia suspensions at −80 °C with 30% glycerol until they were needed for the inoculation of durum wheat plants.

### 2.3. Greenhouse Conditioning and Design of Climate Environments

The plants of three selected durum wheat accessions were grown in greenhouses with full environmental control of the temperature and [$CO_2$]. To establish these weather and

[$CO_2$] conditions, the greenhouses were equipped with air conditioning and dehumidification systems, as well as $CO_2$ supply circuits, all controlled by temperature, humidity, and $CO_2$ sensors, with a fully automated $CO_2$ injection process, to maintain the $CO_2$ target levels (Sysclima, version 9.4, INTA CROP TECHNOLOGY S.L., Murcia, Spain). The established weather conditions were designed to resemble a standard spring day, which is the expected growth period of *Z. tritici* in the wheat growing area of Córdoba.

As average temperatures may not always be an accurate predictor of the potential for infection [31], this study was performed using a variation of temperatures throughout the day, reaching an established maximum and minimum value. Thus, for the baseline conditions, the maximum and minimum temperature values were obtained from the nearest meteorological station, located in Córdoba and belonging to the Spanish State Meteorological Agency, with average values of 24 °C and 10 °C, respectively. Likewise, the [$CO_2$] value was set at around 420–450 ppm, the level currently observed outdoors. Moreover, in order to define the weather conditions for the far future period (2070–2099), the Representative Concentration Pathway RCP8.5 and an ensemble of five climate models (GFDL-CM3, GISS-E2-R, HadGEM2-ES, MIROC5, and MPI-ESM-MR) were used, resulting in average maximum and minimum temperatures of around 30 °C and 15 °C, respectively, and average [$CO_2$] of around 620–650 ppm.

Five sets of plants from three durum wheat accessions were exposed to three different environments, each in separate greenhouses, so as to assess *Z. tritici* infection. Under baseline conditions (environment B), the plants were exposed to a maximum temperature of 24 °C and [$CO_2$] around 420–450 ppm. For the far future scenario, two possible environments were established: under increasing temperature (environment 1), the plants were exposed to a maximum temperature of 30 °C and [$CO_2$] of around 420–450 ppm, and under increasing temperature and [$CO_2$] (environment 2), the plants were exposed to a maximum temperature of 30 °C and elevated [$CO_2$] of around 620–650 ppm.

One set of plants was grown, incubated, and maintained for evaluation under baseline weather conditions (set SB), and four sets of plants were grown under far future weather conditions: two sets in environment 1 and two sets in environment 2. As high temperatures could critically affect the penetration success of *Z. tritici* into the host tissue, two out of four sets of plants grown under far future conditions were inoculated and incubated under baseline weather conditions, before being returned to their far future conditions (sets S1 and S2, respectively). The other two sets of plants were grown, inoculated, incubated, and maintained for evaluation under their respective far future weather conditions (sets S1G and S2G, respectively).

*2.4. Inoculation Assays*

Seeds of the three selected durum wheat accessions were sown in 8 × 7 × 7 cm pots containing a mix (1:1, $v/v$) of commercial compost and sand. The pots were placed in trays and incubated in a growth chamber at 21 °C for a 14 h photoperiod to germinate the plants for 6 days, and then the seedlings were transferred to different greenhouses with diverse weather conditions (environments B, 1, and 2) for 15 days until the third leaf was completely unfolded. Meanwhile, fresh spores of *Z. tritici* were obtained from the spore suspension stored at −80 °C, as previously described by Porras et al. [24]. Thus, a spore suspension was prepared with distilled water and Tween-20 (0.1%) and adjusted to $10^7$ spores $mL^{-1}$. Then, a total of 180 plants (12 per accession and plant set) were inoculated with 7.5 mL per plant of the prepared spore solution of *Z. tritici*, using a hand sprayer until the solution ran off the leaves. Once the leaves were totally dry, the plants were sealed in clear plastic bags to provide 100% relative humidity (RH) for the disease incubation for 48 h. Three sets of plants (SB, S1, and S2) were inoculated and incubated in environment B, while two sets of plants were inoculated and incubated in their corresponding environments 1 or 2 (S1G and S2G, respectively). Finally, the plastic bags were removed, and the plants were kept in their respective environments for 21 days for the subsequent macroscopic and

microscopic evaluations. The macroscopic and microscopic experiments were performed three times each.

### 2.5. Assessment of Macroscopic Components of Resistance

In order to precisely assess macroscopic disease symptoms of STB in wheat plants and those possible changes in the wheat—*Z. tritici* interactions derived from increasing temperature and [$CO_2$], image-based analysis was developed in this study. Several A4 pages that contained a list of sample names according to each accession and plant were printed as templates, as described by Stewart et al. [27]. Each template contained fixed reference points used to set the image scale and boxes within which to mount the leaves. Each box contained the sample name in text as well as encoded as a QR code. Then, the third leaf of 6 plants per accession, plant set, and replication were cut at 21 days post inoculation (dpi), attached to these A4 templates and stored at 4 °C for 2 to 3 days with absorbent paper placed between each sheet of leaves and pressed with approximately 5 kg. Once the leaves were flattened, the templates were digitally scanned at a resolution of 1200 dots per inch using a flatbed scanner (Canon CanoScan LiDE 400, Tokyo, Japan).

Images were analyzed using software ImageJ (version 1.52a) (Wayne Rasband, NIH, MD, USA) [52] using the macro instructions first described by Stewart et al. [27] and later modified by Karisto et al. [26]. The maximum length of leaf area scanned in each box was 17 cm. For each leaf, the following parameters were automatically recorded from the scanned image: total leaf area, necrotic and chlorotic leaf area, number of pycnidia, and their positions on the leaf. Despite obtaining a high efficiency in the identification of pycnidia presented in *Z. tritici* lesions, some of these pycnidia were not counted by the software, which were then manually annotated to increase the accuracy of the measurements. Thus, we calculated the percentage of leaf area covered by lesions (PLACL), the frequency of pycnidia per unit of lesion area (Pyc/lesion), and the frequency of pycnidia per unit of leaf area (Pyc/leaf).

### 2.6. Assessment of Microscopic Components of Resistance

The central leaf segments (~6 cm) of the inoculated third leaves were cut in 3 leaves per accession, plant set, and replication at 4 and 21 dpi. The samples were processed as described in Shetty et al. [53] and then examined using a Nikon microscope (Nikon, Tokyo, Japan). The samples were cleared on filter paper saturated with a mixture of absolute ethanol/glacial acetic acid (3:1, *v/v*) for 24–48 h. The leaves were then transferred to filter paper saturated with lactoglycerol (lactic acid/glycerol/water, 1:1:1, *v/v*), where they were stored until examination. For localization of the fungal structures, the leaves were stained with 0.1% Evans blue (Sigma-Aldrich, St. Louis, MO, USA) in lactoglycerol. In the samples collected at 4 dpi, 400 total spores were observed and classified as follows: spores leading to a stomatal penetration (SP), spores leading to a direct penetration (DP), and spores without penetration (NP) [28,29]. Only the germinated spores were counted. In the samples collected at 21 dpi, 450 total fungal stages were observed and classified as follows: non-colonized stomata (NCS), colonized stomata but not yet transformed into pycnidia (CS), and colonized stomata transformed into pycnidia (Pyc) [28,29]. Both spores and fungal stages of development were photographed using a Nikon DS-Fi1 camera (Nikon, Tokyo, Japan).

### 2.7. Statistical Analysis

The experimental design was developed as randomized blocks. Macroscopic and microscopic parameters whose data did not achieve normality and homogeneity requirements among the different environments for each accession were transformed for statistical analysis with the one way ANOVA test, and transformed back for presentation. However, the parameters whose data could not achieve those requirements using transformations were analyzed through a nonparametric Kruskal−Wallis test. Thus, data from the macroscopic parameter PLACL were transformed according to the formula $y = \sqrt{(x)}$ in the studied ac-

cessions, and were analyzed using one way ANOVA and LSD (Least Significant Difference) tests. However, data from the other two macroscopic parameters, Pyc/lesion and Pyc/leaf, were analyzed using the Kruskal−Wallis test.

In terms of microscopic parameters, data were analyzed using one way ANOVA and Duncan tests for the three selected accessions. However, data from microscopic spore development stages of SP and DP in accessions 'Sy Leonardo' and 'RGT Rumbadur', coupled with data of SP in accession 'LG Origen', were transformed according to the formula $y = \log(x)$. In addition, data from NP in accessions 'Sy Leonardo' and 'RGT Rumbadur' were transformed according to the formula $y = \arcsin(\sqrt{(x/100)})$. Finally, data from fungal development NCS in 'Sy Leonardo' were transformed according to the formula $y = \log(x)$, whereas data from Pyc in 'LG Origen' were transformed according to the formula $y = \arcsin(\sqrt{(x/100)})$. Data processing, statistical analyses, and figure design were carried out using R (version 3.5.0) [54] and ImageJ (version 1.52a) [52] software.

## 3. Results

### 3.1. Macroscopic Components of Resistance to Z. tritici Infection under Climate Change Conditions

The automated analysis of *Z. tritici*-infected leaves carried out in this study led to precisely assessing the differences in macroscopic disease symptoms in durum wheat accessions exposed to diverse weather conditions (Figure 1 and Figure S1). Thus, we evaluated three components of STB infection, such as percentage of leaf area covered by lesions (PLACL), frequency of pycnidia per unit of lesion area (Pyc/lesion), and frequency of pycnidia per unit of leaf area (Pyc/leaf) (Table 1).

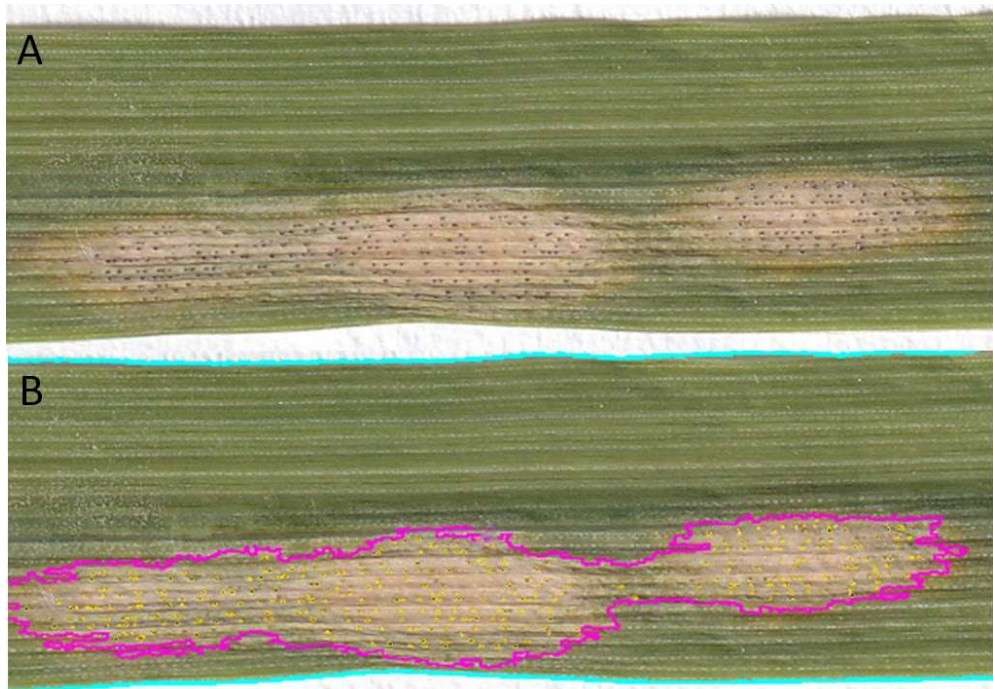

**Figure 1.** Output example of *Septoria tritici* blotch (STB) image analysis developed on selected durum wheat accessions. (**A**) Original leaf; (**B**) Analyzed leaf in which leaf area, lesion area and pycnidia were selected in blue, purple and yellow colors, respectively.

**Table 1.** Macroscopic image analysis of *Z. tritici* infection in three selected durum wheat accessions under baseline and climate change environments after 21 dpi [1].

| Accession | Environmental Set | PLACL (%) | Pyc/Lesion (Pyc/cm$^2$) | Pyc/Leaf (Pyc/cm$^2$) |
|---|---|---|---|---|
| Sy Leonardo | SB | 37.48 (6.11 ± 0.09) a | 601.27 ± 33.09 a | 222.77 ± 10.81 a |
| | S1 | 21.71 (4.61 ± 0.17) b | 226.00 ± 27.66 b | 47.01 ± 5.81 b |
| | S2 | 23.29 (4.79 ± 0.14) b | 266.08 ± 33.85 b | 61.81 ± 8.67 b |
| | S1G | 5.12 (2.21 ± 0.12) d | 105.05 ± 18.32 c | 4.43 ± 0.63 c |
| | S2G | 9.24 (2.97 ± 0.16) c | 139.21 ± 18.75 c | 11.92 ± 1.83 c |
| LG Origen | SB | 9.09 (2.94 ± 0.16) a | 70.83 ± 23.85 a | 4.63 ± 0.91 a |
| | S1 | 1.61 (1.17 ± 0.12) b | 82.05 ± 32.36 ab | 1.43 ± 0.52 b |
| | S2 | 2.19 (1.36 ± 0.14) b | 47.91 ± 20.67 ab | 1.71 ± 0.87 b |
| | S1G | 0.73 (0.56 ± 0.16) c | 2.00 ± 1.70 b | 0.05 ± 0.37 b |
| | S2G | 1.75 (1.26 ± 0.10) b | 44.82 ± 24.46 b | 0.83 ± 0.49 b |
| RGT Rumbadur | SB | 8.88 (2.95 ± 0.10) a | 12.59 ± 2.11 ab | 1.18 ± 0.21 a |
| | S1 | 5.17 (2.22 ± 0.12) b | 12.41 ± 1.65 a | 0.60 ± 0.08 a |
| | S2 | 8.67 (2.88 ± 0.15) a | 7.60 ± 1.06 ab | 0.66 ± 0.12 a |
| | S1G | 1.76 (1.25 ± 0.10) c | 10.44 ± 2.56 ab | 0.22 ± 0.06 b |
| | S2G | 2.42 (1.51 ± 0.09) c | 5.49 ± 1.38 b | 0.13 ± 0.03 b |

[1] Values are mean ± standard error for six leaves evaluated for each accession and environmental set in three different experiments. Transformed data ± standard error are shown in parentheses. Data with the same letter within an accession and column are not statistically different (LSD and Kruskal−Wallis tests, *p* < 0.05). PLACL, percentage of leaf area covered by lesions; Pyc/lesion, frequency of pycnidia per unit of lesion area; Pyc/leaf, frequency of pycnidia per unit of leaf area. SB: plants grown, inoculated, incubated and maintained for evaluation under baseline weather conditions (24 °C and [CO$_2$] around 420–450 ppm). S1 and S2: plants inoculated and incubated under baseline weather conditions, and then maintained for evaluation under far future weather conditions (S1, 30 °C and [CO$_2$] around 420–450 ppm; S2, 30 °C and elevated [CO$_2$] around 620–650 ppm). S1G and S2G: plants grown, inoculated, incubated, and maintained for evaluation under far future weather conditions (S1G, 30 °C, and [CO$_2$] around 420–450 ppm; S2G, 30 °C, and elevated [CO$_2$] around 620–650 ppm).

The susceptible accession 'Sy Leonardo' showed the greatest differences in macroscopic parameter values, with the SB plants expressing higher numbers in comparison with the other plant sets for the three studied parameters. Thus, it showed statistically lower PLACL values for future weather conditions, i.e., S1 and S2 plants (21.71% and 23.29%, respectively) in comparison with SB plants (37.48%). In addition, S1G and S2G plants, which were grown, inoculated, and incubated under environments 1 and 2, respectively, expressed a dramatic reduction in the lesion area caused by *Z. tritici*, showing PLACL values of 5.12% and 9.24%, respectively. This same pattern of reduced values across plant sets also occurred for the Pyc/lesion and Pyc/leaf parameters. Thus, for Pyc/lesion, the S1 and S2 plants expressed acutely reduced values (226.0 and 266.08 Pyc/cm$^2$, respectively) followed by S1G and S2G (105.05 and 139.21 Pyc/cm$^2$, respectively) compared with the SB plants (601.27 Pyc/cm$^2$). Similarly, for the Pyc/leaf parameter, values ranged from 222.77 Pyc/cm$^2$ in SB plants to 61.81 Pyc/cm$^2$ and 47.01 Pyc/cm$^2$ in S2 and S1 plants, respectively, with the values in S2G (11.92 Pyc/cm$^2$) and S1G (4.43 Pyc/cm$^2$) plants being severely reduced.

The moderately resistant accession 'LG Origen' showed statistically lower PLACL values in all plant sets exposed to future weather conditions in comparison with SB plants (9.09%). In this accession, the reduction across plant sets was more pronounced than in 'Sy Leonardo', showing values ranging from 2.19% for S2 plants to 0.73% in S1G plants. For Pyc/lesion, S1 and S2 plants expressed non-significantly different values (82.05 and 47.91 Pyc/cm$^2$, respectively) in comparison with SB plants (70.83 Pyc/cm$^2$), whereas the S1G and S2G plants showed statistically reduced values (2.00 and 44.82 Pyc/cm$^2$, respectively). In contrast, all plant sets exposed to future weather conditions showed relevantly lower values for Pyc/leaf in comparison with SB plants (4.63 Pyc/cm$^2$), with these values being non-statistically significant among them.

Despite resistant accession 'RGT Rumbadur' showed similar PLACL values in SB plants in comparison to 'LG Origen', the observed values for the other plant sets were

not as greatly reduced as for 'LG Origen'. In fact, 'RGT Rumbadur' showed statistically similar values in S2 plants (8.67%) and SB plants (8.88%), followed by significantly reduced values in the other plant sets. Moreover, this accession developed Pyc/lesion values ranging from 12.59 Pyc/cm² in SB plants to 5.49 Pyc/cm² in S2G plants. Lastly, 'RGT Rumbadur' developed the lowest Pyc/leaf values for SB plants (1.18 Pyc/cm²) of the three studied accessions, showing similar values for S1 and S2 plants (0.60 and 0.66 Pyc/cm², respectively), whereas S1G and S2G plants expressed statistically lower values (0.22 and 0.13 Pyc/cm², respectively).

### 3.2. Microscopic Components of Resistance to Z. tritici Infection under Climate Change Conditions

Different stages of spores (SP, spores leading to a stomatal penetration; DP, spores leading to a direct penetration; NP, spores without penetration) and fungal development (NCS, non-colonized stomata; CS, colonized stomata but not yet transformed into pycnidia; Pyc, colonized stomata transformed into pycnidia) were identified during the microscopic evaluation of *Z. tritici* infection at 4 and 21 dpi, respectively (Figure 2), and then analyzed as percentages (Figures 3 and 4, Supplementary Table S1).

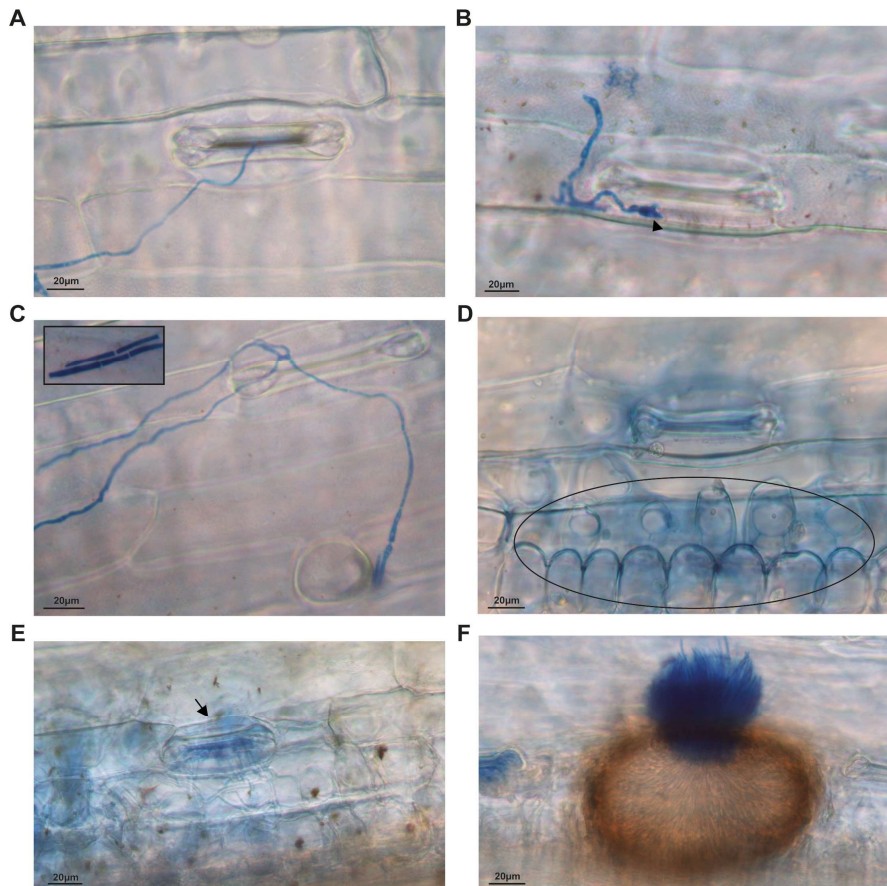

**Figure 2.** Microscopic observation of spore germination, growth, and penetration attempts of *Z. tritici* at 4 dpi (**A**–**C**), and mesophyll colonization and pycnidium structure at 21 dpi (**D**–**F**) are classified as: (**A**) spores leading to a stomatal penetration (SP) and (**B**) spores leading to a direct penetration (DP). Appressorium-like structure formed by an infectious germ tube between stomatal guard cells and epidermal cells over stomata (arrowhead); (**C**) spores without penetration (NP) with non-germinated spore in small square; (**D**) non-colonized stomata (NCS). Longitudinal intercellular growth of infection hyphae growing around mesophyll cells (selected area); (**E**) colonized stomata but not yet transformed into pycnidia (CS). Abundant hyphal growth within the substomatal cavity (arrow); (**F**) colonized stomata transformed into pycnidia (Pyc).



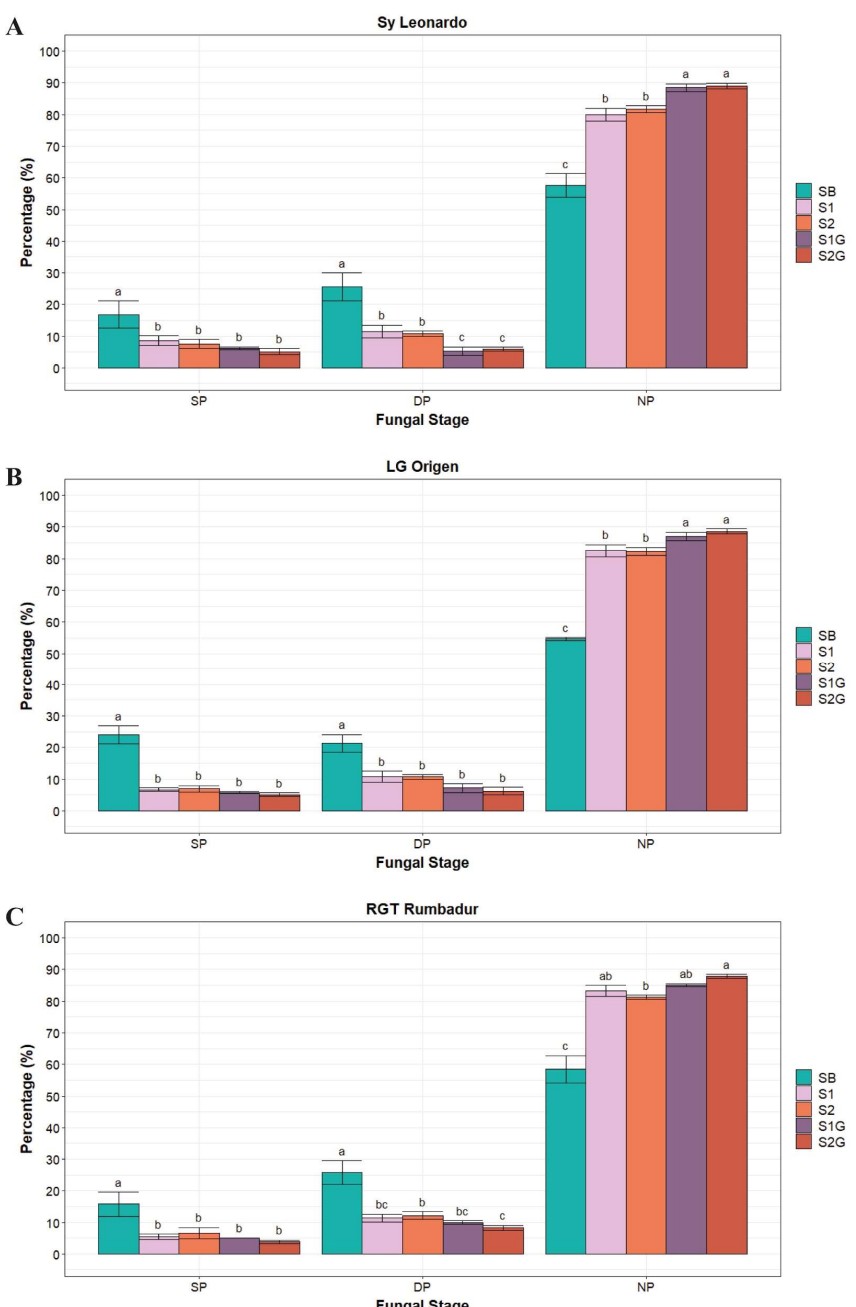

**Figure 3.** Microscopic stages of spores of *Z. tritici* presented as mean percentages in three selected durum wheat accessions (**A**) 'Sy Leonardo', (**B**) 'LG Origen', and (**C**) 'RGT Rumbadur' under baseline (SB) and climate change environments (S1, S2, S1G, and S2G) after 4 dpi. Error bars represent the standard error calculated from three independent experiments. Data with the same letter within a fungal stage and accession are not significantly different (Duncan test, $p < 0.05$). SP, spores leading to a stomatal penetration; DP, spores leading to a direct penetration; NP, spores without penetration. SB: plants grown, inoculated, incubated, and maintained for evaluation under baseline weather conditions (24 °C and [$CO_2$] around 420–450 ppm). S1 and S2: plants inoculated and incubated under baseline weather conditions, and then maintained for evaluation under far future weather conditions (S1, 30 °C, and [$CO_2$] around 420–450 ppm; S2, 30 °C, and elevated [$CO_2$] around 620–650 ppm). S1G and S2G: plants grown, inoculated, incubated, and maintained for evaluation under far future weather conditions (S1G, 30 °C, and [$CO_2$] around 420–450 ppm; S2G, 30 °C, and elevated [$CO_2$] around 620–650 ppm).

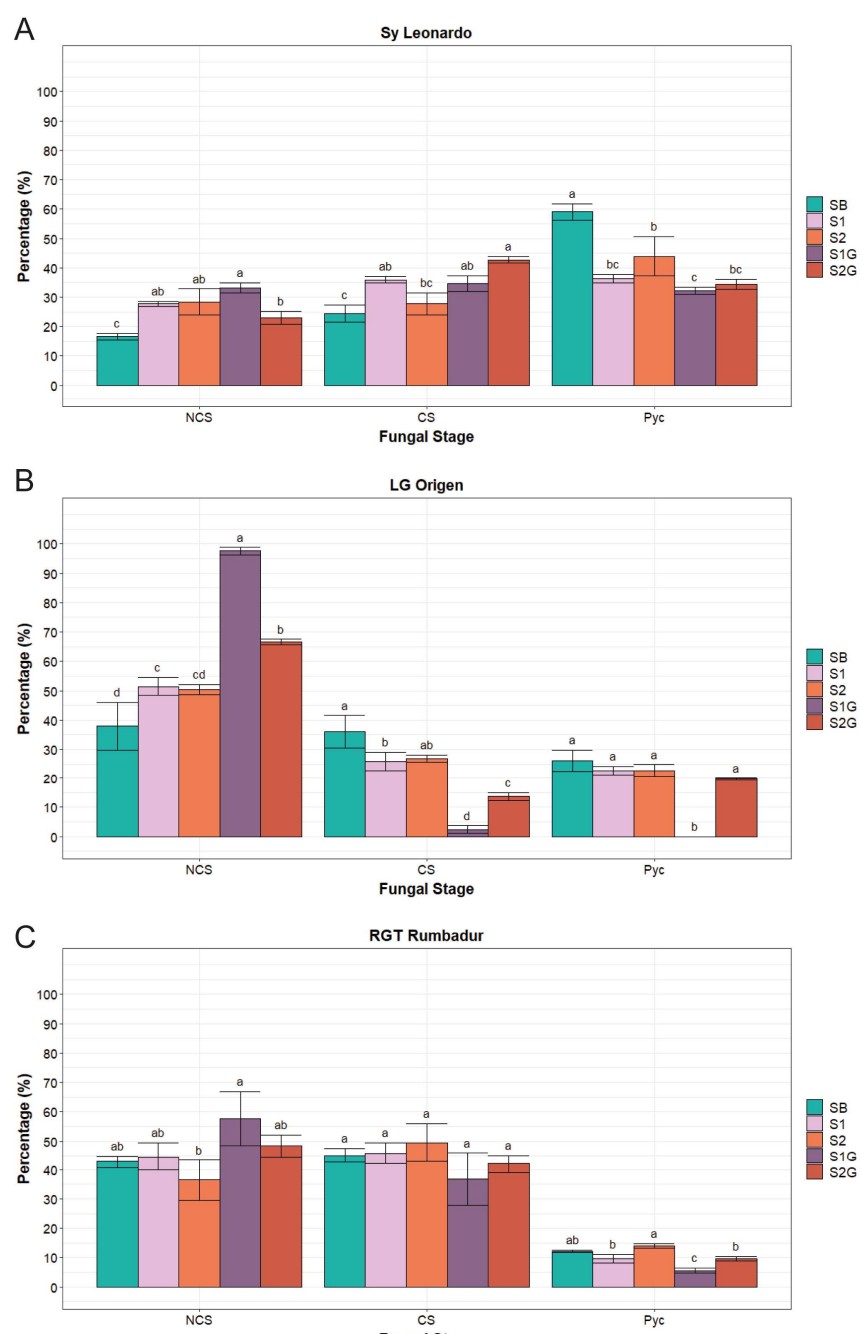

**Figure 4.** Microscopic stages of fungal development of *Z. tritici* presented as mean percentages in three selected durum wheat accessions (**A**) 'Sy Leonardo', (**B**) 'LG Origen', and (**C**) 'RGT Rumbadur' under baseline (SB) and climate change environments (S1, S2, S1G, and S2G) after 21 dpi. Error bars represent the standard error calculated from three independent experiments. Data with the same letter within a fungal stage and accession are not significantly different (Duncan test, $p < 0.05$). NCS, non-colonized stomata; CS, colonized stomata but not yet transformed into pycnidia; Pyc, colonized stomata transformed into pycnidia. SB: plants grown, inoculated, incubated, and maintained for evaluation under baseline weather conditions (24 °C and [$CO_2$] around 420–450 ppm). S1 and S2: plants inoculated and incubated under baseline weather conditions, and then maintained for evaluation under far future weather conditions (S1, 30 °C, and [$CO_2$] around 420–450 ppm; S2, 30 °C, and elevated [$CO_2$] around 620–650 ppm). S1G and S2G: plants grown, inoculated, incubated, and maintained for evaluation under far future weather conditions (S1G, 30 °C, and [$CO_2$] around 420–450 ppm; S2G, 30 °C, and elevated [$CO_2$] around 620–650 ppm).

The microscopic results of the three selected accessions showed similar distribution patterns in the percentages of SP, DP, and NP stages observed at 4 dpi (Figure 3 and Table S1). Accession 'Sy Leonardo' expressed statistically higher values in SB plants for SP (16.85%) and DP (25.57%) stages in comparison with the rest of plant sets for both stages (Figure 3A). Thus, SP values ranged from 8.48% in S1 plants to 5.14% in S2G plants, whereas DP values ranged from 11.48% in S1 plants to 5.26% in S1G plants, with the last and the S2G plant set being statistically different from the S1 and S2 plants. In contrast, SB plants presented a reduced value for the NP stage in comparison with the other sets, showing a percentage far from the S1 and S2 plants (80.04% and 81.65%, respectively) and even more distant from S1G and S2G plants (88.52% and 89.00%, respectively).

In the 'LG Origen' accession, the SB plants differed significantly compared with the other sets for the SP and DP stages, showing elevated percentages of 23.98% and 21.34%, respectively (Figure 3B). Plant sets exposed to future weather conditions did not have statistical differences among them for both stages, with their values ranging from 5.13% in S2G plants to 7.02% in S2 plants for the SP stage, and from 6.22% in S2G to 10.77% in S1 for the DP stage. The NP stage followed the same trend as in 'Sy Leonardo', with the SB plants showing the lowest value (54.68%), while the other plant sets presented higher percentages that were statistically different. Thus, the S1 and S2 plants expressed similar numbers (82.57% and 82.29%, respectively), showing these two sets had statistically different values than the S1G and S2G plants (87.06% and 88.65%, respectively).

The accession 'RGT Rumbadur' showed a similar pattern for the SP stage as in the other two accessions, with the SB value (15.75%) being statistically different from the rest, and the data ranging from 3.83% in S2G plants to 6.62% in S2 plants (Figure 3C). For the DP stage, the plants exposed to environment B also expressed the greatest value (25.80%), showing relevant differences with the other plant sets, which ranged from 8.27% in S2G plants to 12.15% in S2 plants, with only these two sets being statistically different between them. Lastly, plant sets exposed to future weather conditions showed the highest NP values, which were non-statistically different among them, except the S2 (81.23%) and S2G plants (87.90%), whereas SB plants expressed a significantly reduced NP value (58.45%).

Concerning the percentages of fungal stages of development obtained at 21 dpi (Figure 4, Table S1), susceptible accession 'Sy Leonardo' expressed relevantly higher NCS values in plant sets exposed to future weather conditions in comparison with SB plants (16.59%), showing non-statistical differences among them, except for the S1G (33.19%) and S2G (23.01%) plants (Figure 4A). For the CS stage, SB plants also showed the lowest value (24.37%), followed by the S2 (27.78%), S1G (34.66%), S1 (35.93%), and S2G (42.71%) plant sets. However, in the Pyc stage, the SB plants developed the statistically highest value (59.04%) in comparison with the plants exposed to future weather conditions, which showed values ranging from 43.85% in S2 plants to 32.15% in S1G plants, with these two sets being relevantly different between them.

In the 'LG Origen' accession, the S1G plant set showed an outstanding value in the NCS stage (97.56%), and it was significantly higher than the S2G (66.52%), S1 (51.45%), S2 (50.37%), and SB (37.90%) plant sets (Figure 4B). In contrast, for the CS stage, the S1G plant set expressed the significantly lowest value (2.44%), whereas the other plant sets showed values ranging from 13.70% in S2G plants, 25.85% in S1, and 35.98% in SB plants, with these sets also being statistically different. Then, considering the Pyc stage, the S1G plants also developed the statistically lowest score, expressing a null value (0.00%), whereas the other plant sets showed similar values, ranging from 19.78% in S2G plants to 26.12% in SB plants.

The 'RGT Rumbadur' accession developed the slightest differences among the plant sets for the three fungal stages studied (Figure 4C), expressing NCS values ranging from 36.45% in S2 plants to 57.60% in S1G plants, with only these two sets being statistically different between them. For the CS stage, all of the sets were non-statistically different among them. Lastly, the 'RGT Rumbadur' accession, considered to be resistant, developed reduced Pyc values in all plant sets, outstanding SB and S2 plant sets with 12.15% and 14.07% values, respectively, followed by S1 (9.62%) and S2G plants (9.61%), which were

both statistically different compared with the S2 plants. The lowest Pyc value was observed in S1G plants (5.57%), which was different to the other plant sets.

## 4. Discussion

*Zymoseptoria tritici* has become a serious threat for wheat cultivation worldwide thanks to its speciation to diverse agro-ecosystems [8], which would imply an uncertain scenario for the wheat—*Z. tritici* interactions in the context of future climate change, especially for durum wheat species cultivated in regions considered to be hotspots of climate change [48,49]. Although some experimental studies have assessed the effects of increasing temperature [43–45] and [$CO_2$] [46] in the wheat—*Z. tritici* pathosystem, there is scarce knowledge about how the combination of these two abiotic factors, or the diurnal fluctuating temperature as well as the occurrence of heat events, would influence STB disease [31]. Therefore, this study considered the infection of Spanish durum wheat commercial cultivars against *Z. tritici* under diverse weather conditions of increasing (and fluctuating) temperature and [$CO_2$] to elucidate, through macroscopic and microscopic evaluations, the feasible durum wheat—*Z. tritici* interactions. Additionally, in order to assess how the even more frequent heat waves caused by climate change would affect STB disease, expected weather conditions were conducted during both disease establishment and development (S1G and S2G plant sets).

### 4.1. Fungal Penetration Success at Elevated Temperature and [$CO_2$]

In order to assess the capability of *Z. tritici* to penetrate the host plants and cause infection, microscopic observations were conducted at 4 dpi, similar to previous studies carried out in durum wheat cultivars [28,29]. S1 and S2 plants (inoculated and incubated under baseline weather conditions) of the three accessions expressed significantly higher NP values compared with the SB plants. These values could be explained as follows: once the spores germinate, *Z. tritici* hyphae are capable of growing epiphytically on the leaf surface for several days (up to 10) [55]. In addition, S1 and S2 plants, which were returned to environments 1 and 2 after 48 h of disease incubation, respectively, were then exposed to maximum temperatures of 30 °C until observation at 4 dpi. Therefore, this period of exposure could hamper both the stomatal (SP) and direct (DP) penetration of the germinated spores, because temperature seems to have a strong effect on *Z. tritici* spore viability and survival [56–58]. Accordingly, S1G and S2G plants, which were inoculated and incubated under future weather conditions, developed even greater NP percentages for the three accessions, showing an increase in failing to penetrate into the host up to 34% higher compared with the SB plants. Moreover, it seems that events of direct penetration were more restricted in S1G and S2G plants, especially in susceptible accession 'Sy Leonardo', possibly due to elevated temperature reducing the fungal capability of forming appressorium-like structures [53,57]. Considering all of these results, we may suggest, on the one hand, that increasing the temperature up to 30 °C would severely affect the ability of *Z. tritici* to establish the infection process in durum wheat plants, and, on the other hand, that the presence of elevated [$CO_2$] would not modify these fungal patterns. In addition, as the three accessions developed different STB reactions, but expressed similar patterns of fungal penetration, it seems that temperature affected *Z. tritici* development [45,58] more severely than the wheat—plant physiology or wheat—*Z. tritici* interaction at this stage of the infection process [59,60].

### 4.2. Final Disease Development at Elevated Temperature and [$CO_2$]

In order to measure the damage caused by *Z. tritici*, quantification of the final chlorotic and necrotic lesions (PLACL) is usually considered in image-based analyses [26,27,50,61]. In this study, the PLACL values of the plants exposed to future weather conditions in the three studied accessions can be mainly explained regarding the temperature effect reducing spore penetration in microscopic results obtained at 4 dpi. Thus, it is very likely that spores could develop more successful infection sites in S1 and S2 plants rather than in S1G and

S2G plants, even after observations made at 4 dpi [55], demonstrating the reduced PLACL values observed in S1G and S2G plants. In addition, several studies have considered the optimum temperature for disease development under greenhouse conditions to range from 17 to 25 °C [43–45]. Therefore, once disease was established, it is feasible that post inoculation maximum daily temperature in environments 1 and 2 contributed to reducing the subsequent disease progression in plants exposed to these environments until evaluation at 21 dpi. Contrary to this study, Wainshilbaum and Lipps [45] found that wheat plants incubated and evaluated at 29 °C did not show almost any *Z. tritici* symptoms in the final disease evaluation. This is possibly because the temperature range of that study was constantly 29 °C during both the incubation process and disease development, whereas in the present study, the temperature changed throughout the day, highlighting the importance of resembling natural field conditions as well as possible to assess plant–pathogen interactions [31].

Particularly, in the 'LG Origen' accession, considered moderately resistant, the reduction in PLACL values in S1 and S2 plants was greater in proportion compared with 'Sy Leonardo' and 'RGT Rumbadur'. Considering that microscopic parameters observed at 4 dpi and diverse weather conditions were similar for the three accessions, a remarkable post penetration reduction of the disease in 'LG Origen' accession could be suggested. In this sense, elevated temperature has been found to modulate plant resistance against pathogens, increasing or decreasing it in terms of both basal and race-specific resistance [62]. Additionally, increasing temperature is known to upregulate jasmonic acid (JA) synthesis [41], which activates immunity against necrotrophic pathogens [40]. With *Z. tritici* being categorized as a latent necrotrophic pathogen according to recent studies [63,64], it could be possible that partial resistance of 'LG Origen' could be favored at elevated temperatures through JA-induced defense responses.

In addition, it seems that the presence of elevated $[CO_2]$ in environment 2 led plants to express higher PLACL values in comparison with the plants exposed to environment 1. Indeed, some studies support that exposure to elevated $[CO_2]$ increases the disease severity of necrotrophic pathogens such as *Z. tritici* or *Fusarium pseudograminearum*, even in resistant cultivars [38,46]. This enhanced expression of STB disease could be originated from alterations in host physiology produced at high levels of $[CO_2]$ [59,65]. In fact, it is known that elevated $[CO_2]$ enhances the photosynthetic efficiency of plants, especially in $C_3$ plants such as wheat [30], by increasing carbohydrate supply and finally resulting in elevated starch and sugar levels in the leaf tissue [59,66]. Yang et al. [67] also showed that there was enhanced sugar production in the leaves during the symptomatic stage of STB disease. Therefore, it is possible that *Z. tritici* mobilizes leaf sugars for nutritional gain and is favored by the increased levels of sugar presented in leaf tissue at elevated $[CO_2]$. Additionally, there are studies that indicate that exposure to elevated $[CO_2]$ suppresses biosynthesis of stress-induced JA [68], the main phytohormone responsible for plant defense response against necrotrophic pathogens.

Despite these facts, plants exposed to elevated $[CO_2]$ (and temperature) in this study did not express higher *Z. tritici* disease symptoms than plants exposed to baseline conditions, but rather than plants exposed just to elevated temperatures. Therefore, a feasible explanation is that simultaneous exposure to both elevated temperature and $[CO_2]$ forced S2 and S2G plants to develop a unique response during the subsequent *Z. tritici* infection, which could be regulated by antagonistic plant physiological mechanisms, signaling pathways, plant genotypes, pathogen biology, and/or timing and intensity of simultaneous abiotic factors [35,39]. Indeed, differences in PLACL values were not equal among accessions, indicating that diverse environmental conditions could differently influence wheat resistance against *Z. tritici* [61,63]. Thus, it seems that relevant differences in susceptible and moderately resistant accessions arise between S2G and S1G plants, highlighting the beneficial effect of elevated $[CO_2]$ for the fungus when its development was hampered by elevated temperature during the incubation process and host defense responses were not complete. In contrast, resistant accession 'RGT Rumbadur' exposure to simultaneously

elevated temperature and [$CO_2$] counteracted this disease reduction, showing S2 plants to have similar PLACL values to SB plants. Therefore, it is possible that the predicted increase in [$CO_2$] would compromise STB resistant cultivars [46,69], even if exposed to simultaneous elevated temperature during disease progression.

### 4.3. Pycnidia Development at Elevated Temperature and [$CO_2$]

Lesions caused by *Z. tritici* finally develop asexual reproduction structures, called pycnidia [70], which contain asexual pycnidiospores that disperse the disease to other leaves and plants [58]. Therefore, the restriction of pycnidia development can be considered as a key component of resistance against *Z. tritici* [22,25,71]. However, some studies have shown that resistance based on a reduction in pathogen damage to host tissue (indicated by PLACL) can be independent of resistance, which minimizes pathogen reproduction (indicated by Pyc/lesion) [26,61]. Additionally, there is little knowledge about how environmental factors would affect the formation and functionality of these fruiting bodies [71].

In microscopic results, plants of the susceptible accession 'Sy Leonardo' under future weather conditions developed significantly higher levels of non-colonized stomata values (NCS) and a notably reduced proportion of colonized stomata, which finally transformed into pycnidia (Pyc). These values were confirmed macroscopically with a great reduction in Pyc/lesion values in S1 and S2 plants, with this reduction being even more severe in S1G and S2G plants. Therefore, it could be suggested that the elevated maximum temperature of 30 °C affects not only the progression of the disease, but also the ability of the fungus to develop pycnidia [43–45]. However, the reduction in Pyc/lesion values in S1G and S2G plants cannot be explained through microscopic Pyc data values. In this sense, a feasible explanation is that a relevant number of microscopic colonized stomata classified as pycnidia (Pyc stage) were arrested or immature due to elevated temperature [72], which considerably reduces its detection by macroscopic image analysis. This is contrary to the commonly observed pycnidia evolution, where pycnidia become smaller as the density increases on necrotic lesions [73]. Therefore, it is likely that exposure to elevated temperature during the incubation process, as well as during subsequent disease progression, reduce the pycnidia number and may affect pycnidial maturity in S1G and S2G plants in susceptible accessions. Hence, it seems that both *Z. tritici* reproduction and dispersal capability would be severely affected in plants exposed to future weather conditions through reduced pycnidia number and pycnidia size, compromising pycnidiospores production for subsequent disease cycles [25,58,61,74,75]. Lastly, although it seems that elevated [$CO_2$] could slightly improve the formation of pycnidia in S2 and S2G plants in comparison with S1 and S1G plants, respectively, these values were irrelevant, suggesting that elevated [$CO_2$] mainly improved fungal colonization to a certain extent [46,67,68], rather than pycnidia development.

Then, although moderately resistant accession, 'LG Origen' showed reduced Pyc/lesion values in comparison with 'Sy Leonardo'; these values were slightly different among SB plants and plants exposed to future weather conditions. Concretely, only S1G and S2G plants expressed significantly reduced values in comparison to SB plants, confirmed microscopically in higher NCS values and lower CS and Pyc values, singularly in S1G plants. In fact, the supposed reinforced resistance expression of 'LG Origen' under elevated temperature [41,62], coupled with the unfavorable inoculation conditions for a subsequent disease development of S1G plants [43–45], could be reflected in the curiously higher NCS value. Additionally, although S2 and S2G plants expressed similar microscopic Pyc values in comparison with SB plants, they also showed reduced Pyc/lesion values. These results indicate that elevated [$CO_2$] mainly improved the fungal colonization of the leaves instead of developing mature pycnidia in 'LG Origen' accession.

'RGT Rumbadur' accession showed very reduced pycnidia development among diverse weather conditions, which is considered a resistance trait against *Z. tritici* [22,25,26,71] and, obviously, stands out compared with the other two accessions. Focusing on microscopic analysis, NCS and CS values were not statistically different between SB plants and the other plant sets. Indeed, it could be observed that, overall, CS values were higher in

the resistant accession in comparison with the other two evaluated ones, which means the mycelium was generally restricted to the substomatal cavity, and this is supposed to be a specific characteristic of some resistant cultivars [70]. In addition, only S1G plants varied statistically in microscopic Pyc values in comparison with the SB plants, whereas these differences did not occur in the Pyc/lesion parameter. This lack of relevant differences in both microscopic and macroscopic parameters suggests that future weather conditions carried out in this study would barely change pycnidia development in the resistant accession.

Finally, the Pyc/leaf parameter was obtained in order to assess how future weather conditions would affect pycnidia development regarding the whole plant and to identify differences compared with pycnidia development restricted to STB lesions. On this point, differences observed in Pyc/leaf values among plants exposed to diverse weather conditions were almost similar to those obtained in Pyc/lesion values for the three accessions, especially for 'Sy Leonardo'. In fact, the susceptible accession showed statistically similar reduction patterns in plants exposed to future weather conditions for the Pyc/lesion and Pyc/leaf parameters, mainly as a combination of reduced PLACL values, pycnidia number and, probably, pycnidia size. Then, the severe reduction in PLACL values in 'LG Origen' accession led to reduced Pyc/leaf values for plant sets exposed to future weather conditions, showing irrelevant differences among them. Finally, as 'RGT Rumbadur' developed such a low number of pycnidia, it was the accession with fewer changes in Pyc/leaf values through plant sets, mainly due to reduced PLACL values. Additionally, Pyc/leaf values in S2 and S2G plants did not show relevant differences in comparison with S1 and S1G plants. These results suggest that pycnidia development at the leaf level of plants exposed to changing conditions of increasing temperature and [$CO_2$] would vary mainly regarding the lesion surface in which pycnidia could develop, which is mainly affected by temperature, and that variations due to elevated [$CO_2$] would be negligible at this level. Therefore, to detect possible differences under changing environmental conditions, moderately resistant and resistant accessions would require microscopic and specific analysis of lesions caused by *Z. tritici* to a greater extent than susceptible accessions.

## 5. Conclusions

In conclusion, the most important fact in this study is that exposure to elevated maximum temperature alone or in combination with elevated [$CO_2$] not only did not suppress the general defense response in the studied accessions 'LG Origen' (moderately resistant) and 'RGT Rumbadur' (resistant), but also that the disease severity was reduced in these accessions as well as in 'Sy Leonardo' accession (susceptible). This indicates that increasing temperature could mainly affect *Z. tritici* virulence rather than plant physiology, especially in the processes of disease establishment and pycnidia formation and maturation, which would severely hamper the subsequent infection cycles of STB. Concretely, this situation was even worse for *Z. tritici* in the case of plants exposed to climate change conditions during the whole disease process. In contrast, despite the adverse effect of elevated temperature, simultaneous exposure to elevated [$CO_2$] could induce physiological and molecular alterations in the host plant that eventually benefit *Z. tritici* disease development in terms of leaf tissue colonization, especially in the resistant accession, which would threaten resistant cultivars under the predicted [$CO_2$] increase. Finally, it should be noted that for assessment of the climate change effects on wheat—*Z. tritici* interactions, it is essential performing experiments in weather conditions that are as realistic as possible and using both macro and microscopic methods of disease analysis. Despite the progress made in this study, further research should be focus in the combination of three abiotic stresses, increased temperatures, elevated [$CO_2$], and drought, during wheat—*Z. tritici* interactions. In addition, we aim to investigate the cell wall degrading enzymes pattern of *Z. tritici* as well as plant defense responses during the infection process.

**Supplementary Materials:** The following supporting information can be downloaded at: https://www.mdpi.com/article/10.3390/agronomy13102638/s1. Figure S1: Output example of Septoria tritici blotch (STB) image analysis in the studied durum wheat accession Sy Leonardo, LG Origen, and RGT Rumbadur. Upper panels show original leaves. Lower panels show the analyzed leaves in which leaf area, lesion area, and pycnidia were selected in blue, purple and yellow colors, respectively. Numbers indicate selected necrotic areas to be analyzed. Table S1: Microscopic stages of spores and fungal development of the *Z. tritici* infection in three selected durum wheat accessions under baseline and climate change environments.

**Author Contributions:** Conceptualization and supervision, A.P.-d.-L., J.C.S. and I.J.L.; project administration, funding acquisition, and resources, A.P.-d.-L. and J.C.S.; investigation, R.P. and C.M.-R.; methodology, formal analysis, and writing—original draft preparation, data curation, and visualization, R.P.; writing—review and editing, J.C.S., A.P.-d.-L., I.J.L. and C.M.-R. All authors have read and agreed to the published version of the manuscript.

**Funding:** This research was funded by the Agencia Estatal de Investigacion (Spain), grant number PID2020-118650RR-C32, and IFAPA (Spain) grant number AVA23.INV202301.003 (both co-funded by European Regional Development Fund).

**Data Availability Statement:** Not applicable.

**Acknowledgments:** The authors acknowledge the European Social Fund (European Union) and the Agencia Estatal de Investigación (Spain) for PhD grant number BES-C2017-0091.

**Conflicts of Interest:** The authors declare no conflict of interest.

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
