# Peer review of "Characterization of Durum Wheat Resistance against Septoria Tritici Blotch under Climate Change Conditions of Increasing Temperature and CO2 Concentration"

_agronomy, doi:10.3390/agronomy13102638_

Round 1

Reviewer 1 Report

Temperature and [CO2] are the main factors affecting plant development and  disease resistance. In this manuscript, the authors have characterized the response of one susceptible and two resistant durum wheat accessions against STB under different environments in greenhouse assays, simulating the predicted conditions of elevated temperature and [CO2] in the far future period of 2070-2099 for the wheat growing region of Córdoba, Spain. It was found that high temperature could reduce disease incidence, while elevated [CO2] could favored fungal development. These results provide valuable information for understand the effects of temperature and [CO2] on crop growth and disease resistance.

My comments and concerns are as follows:

1, In the section of Results 3.1, here only showed output example of STB image, it is recommended to show disease spot images of three varieties to make it more difficult to visually analyze disease resistance. And supplement the conclusions drawn from this part of the results.

2, In the Table1, it is suggested to add here what do SB, S1, S2, S1G and S2G stand for?

3, Plants are grown in a greenhouse with different concentrations of CO2, and although the set CO2 concentration is fixed, the CO2 concentration changes as the plant grows, so how to control this variable?

4, In the Figure 2, please indicate which figures were observed at 4 dpi and which figures were observed at 21 dpi.

The language is fine.

Author Response

Manuscript ID: agronomy-2663157

Characterization of durum wheat resistance against Septoria tritici blotch under climate change conditions of increasing temperature and [CO2]

We thank the reviewer’s comments. We have thoroughly revised the manuscript to include all their suggestions. Please, find below our detailed responses to the reviewer’s specific questions.

Comments and Suggestions for Authors

Temperature and [CO2] are the main factors affecting plant development and  disease resistance. In this manuscript, the authors have characterized the response of one susceptible and two resistant durum wheat accessions against STB under different environments in greenhouse assays, simulating the predicted conditions of elevated temperature and [CO2] in the far future period of 2070-2099 for the wheat growing region of Córdoba, Spain. It was found that high temperature could reduce disease incidence, while elevated [CO2] could favored fungal development. These results provide valuable information for understand the effects of temperature and [CO2] on crop growth and disease resistance.

My comments and concerns are as follows:

1, In the section of Results 3.1, here only showed output example of STB image, it is recommended to show disease spot images of three varieties to make it more difficult to visually analyze disease resistance. And supplement the conclusions drawn from this part of the results.

Response: Figure S1 has been included in the result section 3.1 in order to address the reviewer’s suggestion.

2, In the Table1, it is suggested to add here what do SB, S1, S2, S1G and S2G stand for?

Response: Suggestion accepted. The footnote in Table 1 has been modified following the reviewer’s suggestions as well as in Figures 3 and 4.

3, Plants are grown in a greenhouse with different concentrations of CO2, and although the set CO2 concentration is fixed, the CO2 concentration changes as the plant grows, so how to control this variable?

Response: The greenhouse used for the development of this study has a CO2 injection and withdrawal system based on the concentrations detected by sensors. Thus, when plants capture CO2 (during the day) and levels go down, the sensors detect this situation and automatically inject CO2 to maintain constant levels. Conversely, when plants expel CO2 during respiration (night), increased levels are detected, and ventilation systems are activated to remove and reduce CO2 levels.

4, In the Figure 2, please indicate which figures were observed at 4 dpi and which figures were observed at 21 dpi.

Response: Suggestion accepted. The footnote in Figure 2 has been modified following the reviewer’s suggestions.

Reviewer 2 Report

The authors propose a manuscript titled “Characterization of durum wheat resistance against Septoria tritici blotch under climate change conditions of increasing temperature and [CO2]”. The article give original information and is well written. The study were conducted on wheat interactions against fungal pathogens, and how will be affected by changes from global climate change. The authors basing on depth knowledge of how predicted increases in temperature and [CO2] will affect especially in durum wheat, which is mainly grown in areas considered to be hotspots of climate change. The study is centred  on the response of one susceptible and two resistant durum wheat accessions against Z. tritici under different environments in greenhouse assays, simulating the predicted conditions of elevated temperature and [CO2] in the far future period wheat growing region of Córdoba, Spain. The experiment suggesting that the temperature is the main abiotic factor modulating the response of this pathosystem. The manuscript deserve of few other crucial informations after which will be able published.

Introduction

Few remarks. Add some references as follow suggested (in bold) because sometimes the concept are already exisisting:

·        However, this pathogen also extends to hot dry climates such as wheat-growing regions of the Mediterranean Basin or North Africa, where durum wheat importance exceeds that of bread wheat due to its common use in the traditional Mediterranean diet [11,12], where lately it is being tested the reintegration of ancient durum varieties of grain more resistant to pathogens [Abenavoli et al. 2021];

·        This situation has hampered the implementation of an efficient strategy to control STB disease, limiting the efficacy of the chemical control [choose reference];

·        Additionally to fungicides application, protection against STB disease has been traditionally achieved through the use of resistant wheat cultivars [choose reference]. Thus, breeding programs emerge as an effective, environmentally sustainable and cost-reducing measure to control this disease in comparison to fungicide control [choose reference].

Reference to be added

Abenavoli, L.; Milanovic, M.; Procopio, A.C.; Spampinato, G.; Maruca, G.; Perrino, E.V.; Mannino, G.C.; Fagoonee, S.; Luzza, F.; Musarella, C.M. Ancient wheats: beneficial effects on insulin resistance. Minerva Medica 2020 Doi: 10.23736/S0026-4806.20.06873-1

2. Materials and Methods

Only two observation.

·     Please give the accession number of durum wheat accessions (T. turgidum ssp. durum) selected from the germplasm collection in Porras et al;

·     Please specify the version of ANOVA used.

3. Results and Discussion

No observation. Weel done. The figures are clear.

4. Conclusions

In the conclusion I suggest to writing two words on the aspect concerning the governance and the perspectives on future studies

Author Response

Manuscript ID: agronomy-2663157Characterization of durum wheat resistance against Septoria tritici blotch under climate change conditions of increasing temperature and [CO2]

We thank the reviewer’s comments. We have thoroughly revised the manuscript to include all their suggestions. Please, find below our detailed responses to the reviewer’s specific questions.

Comments and Suggestions for Authors

The authors propose a manuscript titled “Characterization of durum wheat resistance against Septoria tritici blotch under climate change conditions of increasing temperature and [CO2]”. The article give original information and is well written. The study were conducted on wheat interactions against fungal pathogens, and how will be affected by changes from global climate change. The authors basing on depth knowledge of how predicted increases in temperature and [CO2] will affect especially in durum wheat, which is mainly grown in areas considered to be hotspots of climate change. The study is centred  on the response of one susceptible and two resistant durum wheat accessions against Z. tritici under different environments in greenhouse assays, simulating the predicted conditions of elevated temperature and [CO2] in the far future period wheat growing region of Córdoba, Spain. The experiment suggesting that the temperature is the main abiotic factor modulating the response of this pathosystem. The manuscript deserve of few other crucial informations after which will be able published.

 Introduction

Few remarks. Add some references as follow suggested (in bold) because sometimes the concept are already exisisting:

  • However, this pathogen also extends to hot dry climates such as wheat-growing regions of the Mediterranean Basin or North Africa, where durum wheat importance exceeds that of bread wheat due to its common use in the traditional Mediterranean diet [11,12], where lately it is being tested the reintegration of ancient durum varieties of grain more resistant to pathogens [Abenavoli et al. 2021];
  • This situation has hampered the implementation of an efficient strategy to control STB disease, limiting the efficacy of the chemical control [choose reference];
  • Additionally to fungicides application, protection against STB disease has been traditionally achieved through the use of resistant wheat cultivars [choose reference]. Thus, breeding programs emerge as an effective, environmentally sustainable and cost-reducing measure to control this disease in comparison to fungicide control [choose reference].

Reference to be added

Abenavoli, L.; Milanovic, M.; Procopio, A.C.; Spampinato, G.; Maruca, G.; Perrino, E.V.; Mannino, G.C.; Fagoonee, S.; Luzza, F.; Musarella, C.M. Ancient wheats: beneficial effects on insulin resistance. Minerva Medica 2020 Doi: 10.23736/S0026-4806.20.06873-1

Response: Suggested reference has been included. In addition, references have been relocated and re-enumerated along the manuscript. The former manuscript had 74 references and now there are 75.

Materials and Methods

Only two observation.

  • Please give the accession number of durum wheat accessions (T. turgidum ssp. durum) selected from the germplasm collection in Porras et al;

Response: The durum wheat accessions studied in this study, ‘Sy Leonardo’, ‘LG Origen’, and ‘RGT Rumbadur’, are identified with the name of the durum wheat variety. They are commercial cultivars with no numbers assigned. In fact, in Porras et al., 2021, the supplemental Table S1 shows the same accession names used in this present study.

  • Please specify the version of ANOVA used.

Response: The statistical analysis section of Material and Methods has been modified following the reviewer’s recommendation.

 Results and Discussion

No observation. Well done. The figures are clear.

Conclusions

In the conclusion I suggest to writing two words on the aspect concerning the governance and the perspectives on future studies

Response: The conclusion section has been modified in the manuscript including the reviewer’ suggestions.
